# The Impact of Transportation on the Cortisol Level of Dwarf Rabbits Bred to Animal-Assisted Interventions

**DOI:** 10.3390/ani14050664

**Published:** 2024-02-20

**Authors:** Éva Suba-Bokodi, István Nagy, Marcell Molnár

**Affiliations:** Institute of Animal Husbandry, Kaposvár Campus, Hungarian University of Agriculture and Life Sciences, 40. Guba S. u., 7400 Kaposvár, Hungary; suba-bokodi.eva@phd.uni-mate.hu (É.S.-B.); molnar.marcell@uni-mate.hu (M.M.)

**Keywords:** rabbits, animal-assisted interventions, AAI, stress, transportation, cortisol

## Abstract

**Simple Summary:**

Rabbits’ participation in Animal-Assisted Interventions is increasing. Rabbits are loveable animals, their handling is quick to learn, their body language is simple to read and they can be easily transported into several institutions from kindergarten to palliative care homes. While the positive impact of interventions on humans is indisputable, there is little available information about the effects on animals themselves. Regularly transporting the rabbits to the place of the intervention can be a source of stress, but we assume that they can be trained, and animals can get used to it by offering feed (hay, carrot and apple) and providing appropriate circumstances. Cortisol hormone secretion is initiated as the response to stress and metabolites of cortisol can be isolated in feces samples by laboratory analysis. Using this method we can obtain adequate information about the experienced stress caused the transportation, without disturbing the animals. According to our results, based on the laboratory analysis, repeated transportation causes a significant rise in stress hormone metabolite levels in feces samples regardless of the offered treats during the transport. Those owners who use rabbits for Animal-Assisted Intervention purposes need to take into account that transportation itself is a stressful experience for the animals.

**Abstract:**

(1) Background: the popularity of rabbits has increased during the last decade and become the third most common companion animal in the EU. Rabbits’ participation in Animal-Assisted Interventions (AAIs) is growing. It is highly important to ensure the well-being of the animals in AAIs. Whereas the needs and the advantages of people involved in AAI are becoming more and more evident, the needs of animals are not clearly defined, therefore, it is a great field of inquiry. Animals who are used for AAI need to be transported regularly, which itself might be a source of stress. (2) Methods: the stress of rabbits—caused by transportation—was measured in a non-invasive way: cortisol levels were determined from feces, based on their breakdown products. Eighteen animals were involved in the study. Rabbits experienced a 30 min transportation every second day for two weeks (altogether six times) while 126 samples were collected. (3) Results: rabbits could handle the transportation procedure the first time but subsequently the stress hormone metabolites in feces samples increased regardless of the offered treatments (hay, carrot and apple) during the carriage. (4) Conclusions: those owners who use rabbits for Animal-Assisted Interventions need to take into account that transportation itself is a stressful experience for the animals.

## 1. Introduction

Companion animals have a great impact on human wellbeing. Since the late 1970s it has been scientifically proven that being regularly in contact with pet animals can have positive effects on human’s mental and physical condition [1,2]. Not only the ownership of a pet can cause remarkable positive changes in health and wellbeing, but having brief interactions with companion animals also has the power to decrease the human anxiety level [3].

In the past few decades, the number of companion animals has considerably increased in developed countries. The majority of households keep one or more animals [4]. Several studies have pointed out that the popularity of rabbits has increased rapidly [5,6,7,8]. According to Mikuš et al. [9], the most common companion animal species in the European Union were cats in 2020, followed by dogs [9]. According to the latest 2022 data, the order of the pet animal species did not change but the number of mammals expressively increased in the EU by more than 50% [10].

### 1.1. Animal-Assisted Interactions

The difference between Animal-Assisted Interaction (AAI) and Animal-Assisted Therapy (AAT) must be clarified as these categories are confused in the public consciousness [11]. While the primary goal of AAI is to improve the life quality of the patients through human–animal bonds, in AAT the animals are integral parts of the treatment process. During AAT, the animal and their handler are specially qualified, cooperating with a therapist with credential titles [12]. In contrast, AAI is a less definite human–animal interaction [13].

The first recorded use of animals in therapy was in 1792 in England by William Tuke who began to make efforts to improve the living conditions of mentally ill people [14,15]. To improve the health of people in general, farm animals were used in the earliest decades, while nowadays the most popular therapy animals are dogs [16]. In the last 30 years, other species have also been reported, such as companion horses [17,18], rabbits [19,20,21], cats [22], birds [18], guinea pigs [23] and reptiles [24].

### 1.2. Rabbits in Animal-Assisted Interactions

Rabbits as therapy animals have been used in several social areas, from kindergartens to residential care homes and nursing homes, just as in schools or veterans’ homes [15,25,26]. Rabbits as partner animals in AAIs can function well as an alternative to a dog or a cat, but they need to go through a careful screening process. There are several criteria that must be met in order to use rabbits in AAI programs, according to Granger and Kogan’s [27] and Mallon and Cow’s [28] suggestions. These animals need to be trained:(i)for being transported and for the use of animal transport boxes/cages;(ii)to accept a stranger’s hands-on interaction and to cooperate while being held in their lap for 2 min;(iii)for being stroked or petted by more people at the same time;(iv)to allow being petted in different body parts and areas, including mouth area, teeth, ears and paws, by strangers.

Animals need to be tolerant:(i)with disabled persons in their wheelchairs, crutches, walkers and walking aids for medical purposes, etc.;(ii)to sudden noise, sound and possibly shouting as well.

The rabbits satisfying the above requirements could be involved in various ways in AAI, therefore, they can easily become the children’s favorites [28]. Petting the animal is ideal for developing fine motor skills, learning pet care from an early age, sensitizing children and teaching children responsible pet ownership [27].

AAI could not exist without animals. Therefore, it is of high importance to ensure the health and well-being of the animals in all aspects of AAI. As AAI is beneficial to people through the human–animal bond, it is significantly important not to affect the animals negatively. Whereas the benefits of AAI to people are becoming more and more evident, its advantages for animals are not prominent and hardly measurable. In AAI, the well-being of animals can be harmed in many areas, there is a risk of zoonoses, not only from animals to humans but also from humans to animals, and the physical and mental stress that therapy animals suffer from (even for a short time) can have a negative effect on the well-being of the animal [14]. Menna et al. (2019) do not advise the use of exotic and special pet animals—and rabbits are also included in this category—for AAI purposes. Menna et al., justify their opinion by saying that these species are not closely related to humans on an ethological level [29]. Human–rabbit interactions were measured due to questionaries by Dobos et al., in 2023. In the survey, owners were asked about the interactions and keeping conditions of their pet rabbits. Three principal components were established: amicability, aggression with the owner and aggression with a stranger. Dobos et al., found that individually kept rabbits were considered to be more amicable compared to those that had other rabbits as companions. This human-oriented behavior might be caused by the extra care that a singleton rabbit gets compared to a group-kept one. Dobos et al., also found that large traditional meat rabbit breeds—kept as pet rabbits—are less amicable than dwarf pet rabbit breeds. Even the type of feeding showed an association both with amicability and aggression. Pet rabbit owners often offer muesli-type feed that is calorie-rich and soft. These have negative consequences for the rabbits’ body condition and can cause painfully overgrown teeth which could possibly result in more frequent aggressive responses towards their caregivers [30]. In order to involve rabbits in AAI it is essential for kits that right after birth—while they are in the sensitive period—they become accustomed to the touch of the human hand. This early handling—called imprinting—has a positive impact on stress tolerance throughout the lifetime of the rabbit. Consequently, to ensure the rabbits’ well-being, it is beneficial for therapy rabbit breeders to familiarize kits with human hands and to have prior knowledge about the rabbits’ temperament [31].

### 1.3. Considering Transportation Stress

Transportation of animals, regardless of the species, is a stressful experience [32,33,34], therefore, commercial animals’ transportation processes are regulated by law and must be documented. Specific provisions relating to carrying companion animals are universalized and give ordinary wellbeing instructions [35].

The effects of carriage on dogs are well documented, such as the methodology of observing the dogs’ behavior during transportation [36,37,38]. Specific data, such as stress hormone (cortisol) examinations, are also available [39,40,41,42].

Generally, rabbit transport-related publications are focused on slaughterhouse deliveries in order to improve meat quality [43,44,45]. The European Food Safety Authority (EFSA), panel on Animal Health and Welfare (AHAW) 2022 gave a scientific opinion about rabbits’ container transport’s welfare consequences [46] in order to reduce the animals’ suffering of trauma. As a result of inappropriate animal handling during transportation typical injuries occur, such as hemorrhages, bruises and broken bones. The EFSA and the above publication’s common recommendations are the following:(i)careful rabbit handling,(ii)to prevent the transfer of urine and feces, solid floors of the crates are recommended,(iii)to avoid heat or cold stress, rabbits should travel in their thermal comfort zone. To fulfill this criterion the temperature inside the van should be kept permanent, between 10 °C and 20 °C.(iv)Prolonged hunger (more than 12 h feed withdrawal) causes weight loss and is pernicious to their welfare. In the thermal comfort zone, the total time of food and water delimitation should not exceed 12 h.(v)the crates’ height should be enough to let the rabbits sit in a natural position.

Buil et al. [44] highlight that the critical point of commercial rabbits’ transport to slaughter is the condition of transport while the time of the journey is a less stressful aspect. Therefore, the details of transportation need to be well planned and executed in advance.

Herbel et al. [47] examined the stress response of beagle dogs where 18 animals attended the study and undertook repeated short-distance road transports. Transport cages were placed on the floor of a minibus that had all its rear seats removed. The route took 60 min included approximately 25% city traffic and 75% four-lane motorways. The driver of the car and a second person accompanying the transports had handled the dogs during the four-week accustomization phase before the study and were, thus, familiar to the dogs. The same two persons were also present during control experiments. Our present study was based on a similar design and carried out parallel methodology in several respects (detailed in Section 2).

### 1.4. Non-Invasive Techniques for Analyzing Hormonal Indicators of Stress

The sympathetic-adrenal medullary (SAM) and the hypothalamic-pituitary-adrenal cortex (HPA) systems—the adrenal axes—are activated by environmental stimuli. Mammals produce hormones (catecholamines and glucocorticoids) in order to ensure the body gets readily available energy in an emergency action. By monitoring catecholamines and glucocorticoids the animals’ short-term stress is assessable. The plasma’s glucocorticoid level is increased by the activation of the HPA axis, consequently, analyzing the blood’s stress hormone levels is possible [48,49]. The most common method for assessing hormone levels is a blood test, but this method has many limitations, especially in the diagnostic process of non-domestic animals [50]. The procedure of sampling the animals needs to be undertaken by capturing, clamping and venipuncture, which is a highly stressful situation for them and thus increases stress hormone levels [48,51].

An alternative method for blood sampling is measuring glucocorticoid levels in feces, urine, saliva, or hair in various animals in order to quantify stress [48,50]. Samples collected in a non-invasive way are stress-free methods and, therefore, are highly considerable to fulfill animal welfare criteria [48,50]. Metabolites of glucocorticoids (GCMs) appear in feces, reflect the accumulation of glucocorticoids over several hours and can be collected without disturbing the animal. Thereby, GCM is used in rabbits as an indicator of stress provided in a non-invasive way [51], although, as further bacterial metabolism of the collected feces cortisol/corticosterone metabolites may occur, the sampling and storing conditions are critical [52]. After defecation, the feces samples need to be collected and frozen within 30 min and must be kept under −20 °C until analysis [52].

According to Benedek et al. [53], 24 h after the pressure of stress the decomposition of the hormone cortisol appears in the feces.

Rabbits’ participation in Animal-Assisted Interventions is increasing, however, no recommendation is available about how to use them while animal welfare is guaranteed. As those rabbits that are a service in AAI regularly undertake transportation to get to the plot of the sessions, our present study’s main objective was to investigate the effect of the repeated transportation on the rabbits’ stress level and give general recommendations about how to fulfill animal welfare requirement during the procedures.

We hypothesize that transportation negatively affects the rabbits’ anxiety level, but they can become accustomed to short-lasting trips by regularly undertaking them to attend the procedure. We also hypothesize that different types of feed or treats offered to the rabbits during transportation reduce their anxiety level.

## 2. Materials and Methods

### 2.1. Data of the Involved Rabbits

The study was conducted with 18 dwarf rabbits selected for tameness during seven generations. The selection was based on the rabbit’s Human Approach Test according to Suba-Bokodi’s study [31].

In this examination, ten bucks and eight does were involved. The rabbits were between the ages of 12 and 18 months. 

### 2.2. Housing and Feeding

In the rabbit shed each rabbit was housed individually as presented in Figure 1. The Hungarian and EU legislation in force Decree of the Ministry of Agriculture (32/1999./III. 31./ and 178/2009./XII. 29./) in the “General rules for rabbit does and bucks, and suckling and growing rabbits” gives a clear policy: the rabbits must be housed individually after 12 weeks of age, with the exception of fattening rabbits. The individual cages of our rabbits were placed next to each other and the animals were in visual contact. The pet rabbit cages (95 × 57 × 46 cm) have a corrosion-resistant metal-coated wire mesh (on the top) and a dark plastic bottom. The spacious rabbit cage provided all the necessary things rabbits needed in their cage, like a feeder, a hay container, a hanging water bottle, a separate sleeping area and a rabbit litter box filled with wood pellets (compressed wood shavings and sawdust from untreated wood). To be a house-trained animal, rabbits used the litter box to urinate and defecate. All the accessories were made of plastic except the stainless-steel feeder.

As for feeding, the rabbits received a complete diet (Versele-laga complete cuni adult rabbit) which is an all-in-one pellet to avoid selective feeding behavior and to ensure consumption of all the essential nutrients for optimal health. This feed does not contain a coccidiostat or other additives. In addition to the pellets, the rabbits had free access to the water nipple drinker, were given hay ad libitum and a daily 35–40 g of fresh raw carrot sticks and 30–35 g apple pieces, both from organic farming, as a treat and vitamin complementary. The cages were equipped with gnawing sticks and supplementary mineral blocks as environmental enrichment. Although the animals were housed individually to prevent harm and social stress, the cages were placed side-by-side, enabling them to sniff each other and stay in visual contact. In good weather conditions, the rabbits were grazing for at least two hours in their mobile open-air hutches every second day as presented in Figure 2a,b.

Veterinary checkups, including physical examinations like dental health condition checks and assessment of the appearance of the *Spilopsyllus cuniculi* and Psoroptes, were conducted on all rabbits prior to the transportation examination. All rabbits were vaccinated against myxomatosis and RHD, and none of them were infected with parasites or zoonotic diseases.

### 2.3. The Transportation by Car

For two weeks, every second day the rabbits were taken for a 30 min transport by car at 8:30 AM, imitating the duty of an assisted animal that is working three days per week in AAI and needs to be transported to the intervention’s location. During the examined period, the rabbits were transported on Monday, Wednesday and Friday, while on all the other days they remained at their housing. They were transported in rabbit-sized cages, made of strong plastic and consisting of two parts that are held together with a special fastening system made up of a clasp on the back and two hooks in front. It comes with a plasticized metal door with a safety clip. The upper part has an ergonomic handle to ease carrying and ensure a sturdy grip. Side grids make sure there is air circulating inside. According to the carrier’s direction of use, it is suitable for pets weighing up to 3 kg. The approximate size of the box is 24.5 × 28 × 41.5 cm. The transportation dates were the following: 24 October; 26 October; 28 October; 31 October; 2 November and 4 November 2022. The transportation was carried out by an air-conditioned minibus (Renault Trafic Combi, 2021 model, France) at a temperature of 19 degrees. The placement of the carrier boxes are presented in Figure 3. Each carrier was stabilized on the floor to avoid side slips. Twelve rabbits were transported in the first week and six in the second. While traveling in the car, rabbits had limited access to see each other and the outside environment during the experimental stages. Each trip was preceded by a 10 min familiarization experience when rabbits were permitted to get used to their cages in the non-moving and shaded car. The 30 min (25 km) public road drive consisted of, approximately, 10 min of city traffic and 20 min of a two-line main road. The same route was driven on each occasion. The owner of the rabbits and a second person—the car driver–were present during the transportation.

The methodology was based on the study by Herbel et al. [47] which examined dogs’ short-term transportation. The 18 rabbits were divided into three groups:Six of them were transported three times to examine intermediate long-period stress effects on the animals;Six of them were transported six times to examine prolonged period stress effects on the animals;Six animals formed the control group that were not transported at any time and remained at their housing during the whole examination period.

In order to obtain information on whether the experienced stress is reducible by feeds or treats, three subgroups were formed from the transported animals (four rabbits in each group): no feed, hay and hay+ carrot groups, detailed in Table 1.

### 2.4. Collecting, Storing and Transporting the Feces Samples to Laboratory

Samples were collected seven times from 18 rabbits, so the total sample number of this research was 126. The first samples were collected two days before the examination period in order to obtain information on the individuals starting cortisol levels. The second time sampling was exactly 24 h after the transportation had been carried out as the hormone cortisol’s metabolites appear in feces with a delay [51,53]. The sampling dates were the following: 22 October; 25 October; 27 October; 29 October; 1 November; 3 November and 5 November 2022. The samples were collected as follows:The plastic bottom of the animal’s individual cage and the litter box were cleaned and sterilized with a biocidal product with a spectrum of bactericidal, fungicidal and virucidal effects (SteriClean Farm, active ingredient: sodium hypochlorite solution (0.05%)) daily at 7:30 AM.After restoring all equipment in the cage, the litter box was filled with wood pellets again.At 8:30 AM, if feces appeared in the litter box, they were removed immediately.After 8.30 AM, from the first feces of the animals, samples were collected using sterile gloves for each individual. Every 15 min the conductor of the examination (who is the owner of the rabbits—first author) checked the litter boxes and collected the new samples.All the samples were labeled (with the rabbit’s number and sampling date) and immediately frozen.The samples were kept at −21 °C until they were transported to the laboratory.

The samples were transported in a cold box, that is suitable for vaccine carries, in a frozen condition, on 9 January 2023 to the Department of Diagnostic Laboratory, University of Veterinary Medicine, Budapest, Hungary, where the laboratory examinations were carried out. Cortisol levels were measured from the feces based on their breakdown products according to Benedek et al.’s [53] method. Cortisol was extracted from the feces samples by adding 4 mL of ethanol for 500 mg of feces, followed by vortexing for 3 min. Then, 10 min centrifugation at 2000 rpm was applied. Cortisol-containing supernatant was collected and used as samples.

The concentration of metabolites was determined with a cortisol ELISA kit (DEH3388, Demeditec GmBH, Kiel, Germany), according to the manufacturer’s protocol. Analytical sensitivity 0.38 ng/mL; range 10–800 ng/mL; intra-assay CV < 5%; inter-assay CVs were 5.2 and 7.8% for control 1 and control 2, respectively. Raw cortisol concentration data were translated and expressed as μg/g.

In the groups formed by the number of transportations, the appearance of cortisol concentration, defined by the metabolites in the feces samples (μg/g), was interpreted per day and presented using the following scheme:PRE—collected 2 days before the transport began (22 October),1—collected 24 h after the first transport (25 October),2—collected 24 h after the second transport (27 October),3—collected 24 h after the third transport (29 October),4—collected 24 h after the fourth transport (1 November),5—collected 24 h after the fifth transport (3 November),6—collected 24 h after the sixth transport (5 November).

### 2.5. Statistical Analysis

The cortisol level of the rabbits’ feces was the dependent trait and was analyzed using the Generalized Linear Model (GLM) repeated measures procedure applying the SPPSS software (version 27, IBM-SPSS, Armonk, NY, USA). This procedure takes into account that the cortisol level of every rabbit was measured several times, thus, the repeated measurements belonging to the same animals were not independent. The considered factors were days, sex, times of travel, feed and their interactions: Cortisol = days + sex + times of ravel + feed + (sex × times of ravel) + (sex × feed) +  (times of ravel × feed) + (sex × times of ravel × feed) 

In the case of the significance of any factor or factor interaction (*p* < 0.05), Tukey post hoc tests were performed.

## 3. Results

Table 2 presents the feces cortisol levels of the individuals: the first column shows the rabbit number used for the examination, while the second one shows the different subgroups created by the various feed they received during the transportation, in alphabetical order (detailed in Section 2.3—Material and Methods, The Transportation). From the 3rd to the 9th columns the measured cortisol values are presented—provided by the laboratory—on a daily basis in μg/g. The 4th column is labeled “Pre”. That data represents the rabbits’ individual initial cortisol levels, as they were the first collected samples made two days prior to the examination period in order to obtain information on the animals’ normal cortisol levels. Columns 1st to 6th show the rabbits’ feces cortisol levels in μg/g. The samples were collected 24 h after the transportation was carried out, detailed in Section 2.4. 

According to the data of the 18 dwarf rabbits’ cortisol (in μg/g) levels, as determined by the 126 feces samples, there is no significant difference between the bucks and does. Additionally, there is no significant difference between the additional feeds offered during transportation and the rabbits’ cortisol concentration as determined by their feces metabolites. Hereafter, we continue the analysis of the results only according to transportation and control groups.

The elapsed days’ effect on the rabbits’ stress level is demonstrated in Figure 4.

The detailed statistical data of the elapsed days effect are the subject of Appendix A. The difference is significant (*p* < 0.001) between the following days: Pre–5; 1–4; 1–5 and 5–6.

Table 3 shows the frequency of transportation’s effects (three or six times) on rabbits’ stress. There is a significant difference between the control animals and the three-times (*p* = 0.028) transported ones, and also between the six-times transported animals (*p* = 0.015). According to the data, the stress level is increased by continuous transportation, but there is no significant difference between the three- and six-times transported animals (*p* = 0.871).

Figure 5 shows the estimated marginal means of the cortisol (μg/g) level, determined from the metabolites of feces samples measured after the occasion sessions on a daily basis, according to the transportation’s frequency.

The sample collection—detailed in Table 2—started on 22 October 2022, two days before the first transportation was carried out. Between the groups, there was no significant difference in the cortisol levels determined by its decomposition in the feces samples. The mean feces cortisol levels in the transported rabbits group were established at 6.58 μg/g, while 6.97 μg/g was that of the control group. The total mean of all animals who participated in this study was 6.71 μg/g.

The first week’s transportations were carried out three times: 24, 26 and 28 October 2022. The measured feces cortisol levels after the first occasion were statistically the same as the first samples that had been collected before the transportation began. On the second occasion (26 October), all the feces cortisol concentrations were higher compared to those on the starting date. On the second day of transportation (column 3), the transported rabbits’ cortisol levels were 4.89% higher than the control group’s. The difference had decreased on the third occasion. The tendency of having a gap in cortisol values between the transported animals and those rabbits who remained at their housing can be determined, but none of the days’ differences are significant.

During the first week, 12 rabbits had been transported, by the second week (31 October, 2 and 4 November) 6 of them remained at their housing (group of “three times transported” rabbits, marked by red columns in Figure 5 and the other 6 rabbits were continuously transported. On each occasion, the continuously transported ones had higher cortisol levels, determined from feces samples, compared to the other two groups: the control and the non-transported ones. The transported rabbits’ feces cortisol concentration peak was 157.04 μg/g on the 1st of November (number 5 column) and after that a decreasing tendency was dominant. By the last transportation session, the stress hormone concentration was reduced to almost one-third (64.8 μg/g).

## 4. Discussion

The outcome of this research provided insight into the rabbits’ anxiety levels during transportation. Our hypothesis that transportation negatively affects the rabbits’ stress levels is confirmed.

In order to obtain information about the rabbits’ starting stress levels, sampling from all the rabbits was carried out two days before the transportation started. The total mean of feces cortisol was 6.71 μg/g at the beginning and it was 6.58 μg/g after the first transport. After a day off (rest period), the examination was continued, the rabbits were taken to the same session and the feces samples were collected again. Their feces cortisol concentration rose to 90.85 μg/g.

The control rabbits’ cortisol levels also rose from the second day of the transportation and moved similarly to that of the transported rabbits. This tendency persisted until the end of the research, although a significant difference was found to exist between the animals who remained at their housing during the whole study and the transported ones. The reason for the control rabbits’ cortisol rise has been investigated. The circumstances of housing did not change during the study. The rabbits were kept under thermoneutral conditions and were individually housed since they were 12 weeks old. Their cage was placed in the rabbit stable and had not been moved or changed in position for at least six weeks before this study was carried out. No equipment was changed, and the daily routine was made exactly the same as the animals were getting used to previously. Ad libitum hay, pellet food and water were available for them, just as before the study. However, it must be noted that the rabbits’ housing system and the care of the animals are more similar to that of pet animals than stock rabbits. They receive more stimulus from the environment: they are cleaned up every second day, often touched by their owner and regularly graze in the open-air hutch. The animals were taken care of by their owner who was handling them during the transportation and was present entirely. Before the experimental period, the attendance of the rabbits’ owner meant a positive factor for the animals as they were treated carefully and offered feed as a daily treatment. In contrast, as the owner undertook some of the animals to a stressful event, this attitude toward the owner might have been changed. Because of the significant rise in feces cortisol concentrations in both groups (the control and the transported), we suppose that animals may share their stressful experiences, but in a different way than humans do, in a way that human senses are not able to express or detect. Previous studies proved that in different stressful situations, lambs showed coherent emotional reactivity by alerting their behavioral and physiological responses [54]. Stressful events cause catecholamine and glucocorticoid production through the activation of the sympathetic system and hypothalamic-pituitary-adrenalin axis [55]. However, there is little known about how stressful experiences influence the animals’ relationship with each other, but it is well established that humans develop stronger relationships as a result of such experiences.

To survive, cooperation between the animals may improve social interoperability when animals share negative experiences [56]. In our present study, the control group’s feces cortisol levels rose from the second day of transportation (26 October) to the last occasion (4 November) when it was reduced by more than 80% and established at a level of 17.92 μg/g on the last examined day. Those rabbits that had not been transported and remained at their housing during the whole study also showed increased cortisol levels.

On the basis of the presented data, the conclusion is drawn that one occasion of 30 min road transport in individual boxes under thermoneutral conditions did not have an essential impact on rabbits, and as there was no significant difference between the transported and the control groups, we suppose that the animals were able to cope with the stress caused by the first transport. From the second transportation, the difference did appear between the transported and control group’s feces cortisol levels, although both groups’ stress hormone values rose. There was a significant difference between the control animals and the three-times (*p* = 0.028) transported ones, and also between the six-times transported (*p* = 0.015), but there was no significant difference between the three- and six-times transported ones (*p* = 0.871). All animals’ feces cortisol levels were considerably reduced on the last examined day.

We can conclude that transportation negatively affects the rabbits’ anxiety level but, based on the large-scale reduction that appeared by the 6th session, we can assume that rabbits might be trained for short-term transport, i.e., they can get used to it by regularly undertaking transport to the procedures while careful handling is ensured. To avoid the exhaustion of rabbits that are used for assisted/therapy purposes, we suggest that trainers who are responsible for the animals and ensure their welfare during the interventions take into consideration that transportation itself causes a highly stressful experience. Presumably, rabbits can get used to the transportation because the feces cortisol hormone appearance showed remarkable degradation by the last sampling date in our study, but the two-week period of training was not enough for the rabbits, therefore future studies are needed.

We supposed that different types of feed or treats offered to the rabbits during the transportation could reduce their anxiety level. According to the examined feces cortisol levels collected after the transportation of the rabbits, our hypothesis failed to be proven, although only two animals remained in each subgroup. We could not establish a correlation between the stress level of the transported animals and the offered hay or other various supplementary feed, so our hypothesis must be rejected, but we do suggest further studies to investigate whether the anxiety of the animals is reducible by feed, as it was been observed that some of the rabbits ate while they were at their carrier cages (Figure 6a,b).

## 5. Conclusions

Transportation itself puts pressure on the animals in addition to the stress caused by Animal-Assisted Interventions. Generally, rules of transportation are regulated and adjusted to animal species, but these orders mainly apply to a single visit to the slaughterhouse. However, animals must be transported to AAI sessions regularly, even several times per week. There are only a few available studies about the transport of companion animals, therefore, data obtained on dogs can be the guide with the understanding that the rabbit is a prey animal species and may react differently in stressful situations like transportation. During our study, the rabbits were transported six times in two weeks, for 30 min on each occasion. After the transports, feces samples were collected and analyzed in order to determine the appearance of the stress hormone cortisol’s metabolites. This method gives information about the animals’ experienced stress in a non-invasive way. In our present study, none of the feeds (hay, carrot and apple) had an effect on the animals’ stress; we cannot establish a correlation between the stress level of the transported animals and the offered hay or other various supplementary feed. However, we must note that some of the rabbits ate while they were in their carrier cages, which is why we suggest offering them additional treats and hay to offer a chance to occupy themselves during the transport.

We can conclude that repeated transportations negatively affect rabbits’ stress levels. To avoid the exhaustion of the rabbits that are used for assisted/therapy purposes we suggest that trainers who are responsible for the animals and ensure their welfare during the interventions must take into consideration that transportation itself causes a highly stressful experience. Presumably, rabbits can get used to the transportation, because the feces cortisol hormone appearance showed a significant degradation by the last sampling date in our study, but the two-week period of training was not enough for the rabbits, therefore, future studies are needed.

## Figures and Tables

**Figure 1 animals-14-00664-f001:**
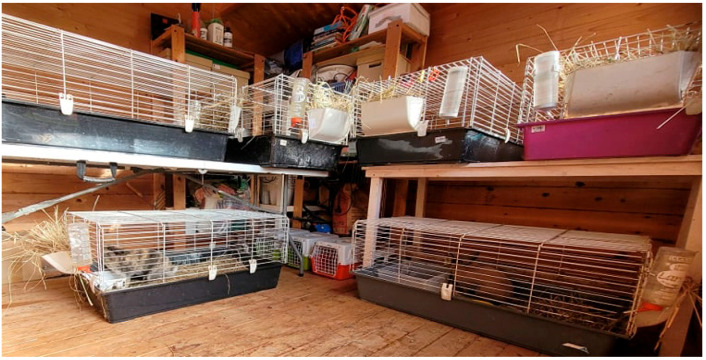
Rabbit shed.

**Figure 2 animals-14-00664-f002:**
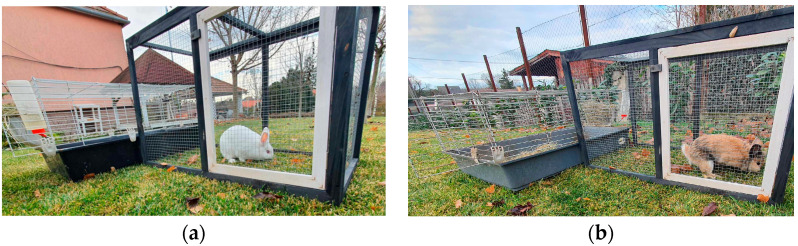
(**a**) Rabbit free grazing. (**b**) Rabbit free grazing.

**Figure 3 animals-14-00664-f003:**
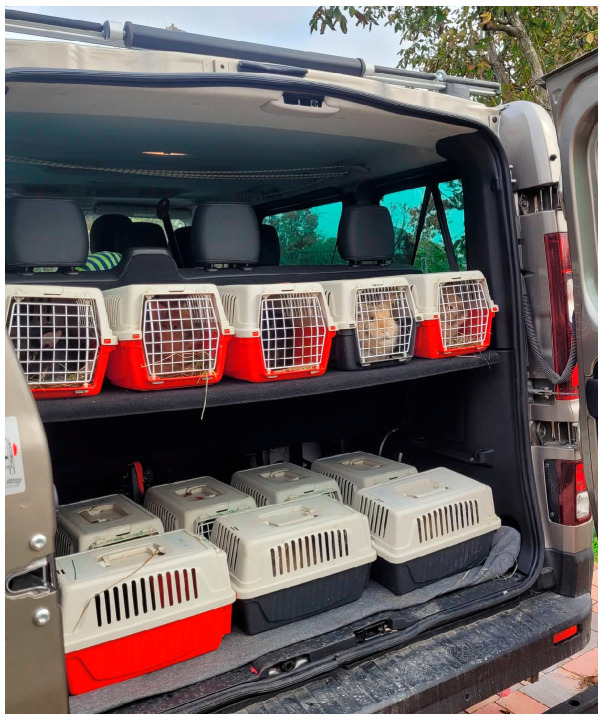
Rabbits in their individual transportation boxes placed in the car.

**Figure 4 animals-14-00664-f004:**
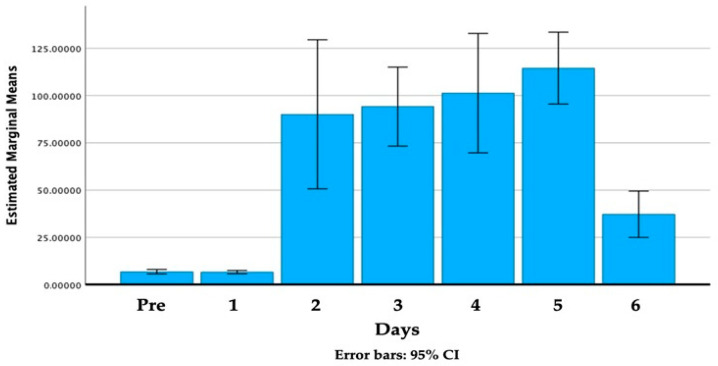
The effect of elapsed days on the rabbits’ stress hormone, cortisol (in μg/g), as determined from the metabolites of all transported animals’ feces samples.

**Figure 5 animals-14-00664-f005:**
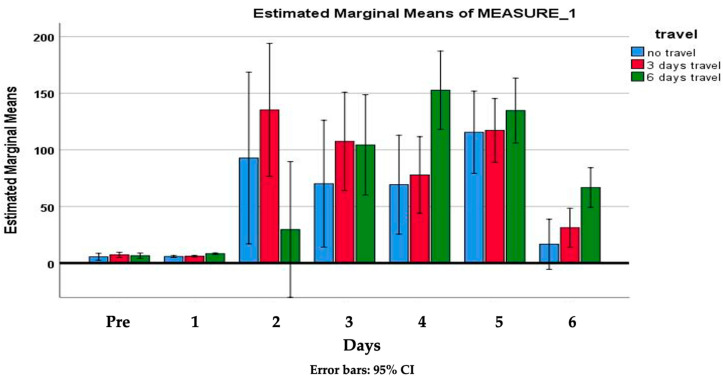
Rabbits’ feces cortisol levels (μg/g) during the examination period, according to the frequency of transportation.

**Figure 6 animals-14-00664-f006:**
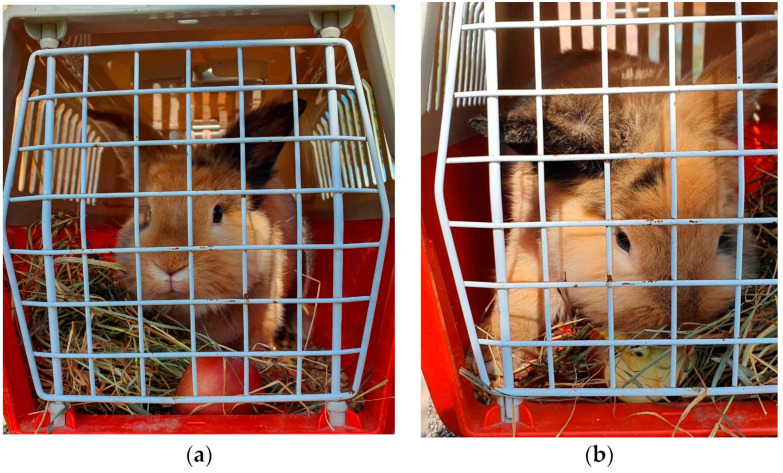
(**a**,**b**) Rabbit in the individual carrier cage eating treats.

**Table 1 animals-14-00664-t001:** Grouping of the animals according to sex, number of transportations and feed.

Rabbit’s Number	Sex	Number of Transportations	Feed *
1	F	6	No feed
2	M	6	No feed
3	M	6	hay
4	M	6	hay
5	M	6	hay + carrot
6	M	6	hay + carrot
7	M	3	No feed
8	F	3	No feed
9	F	3	hay
10	M	3	hay
11	M	3	hay + carrot
12	M	3	hay + carrot
13	F	0	control
14	M	0	control
15	F	0	control
16	F	0	control
17	F	0	control
18	F	0	control

* where: No feed: Rabbits (*n* = 4) that during transportation only wood pellets (used also for housing) were on the ground of the carrier without additional feed (*n* = 4), Hay: Rabbits (*n* = 4) that during transportation wood pellets (used also for housing) were on the ground of the carrier and a handful of hay (given ad libitum while housing) was put in the carrier, Hay + carrot: Rabbits (*n* = 4) that during transportation wood pellets (used also for housing), a handful of hay (given ad libitum while housing) and their daily portion of carrot and apple (approximately 70–75 g) were put in the carrier, Rabbits of the control group (*n* = 6) that had not been transported remained at their housing during the whole examination period.

**Table 2 animals-14-00664-t002:** Measured cortisol concentration from the metabolites of feces samples in (μg/g) pre (column 3), and 24 h after, the transportation (columns 4–9).

Rabbit Number	Times of Transportation	Feed	Feces Cortisol (in μg/g) after Repeated Transportation						
			Pre	Day 1	Day 2	Day 3	Day 4	Day 5	Day 6
1	6	no feed	5.86	10.29	6.01	99.22	121.81	153.51	64.80
2	6	no feed	8.84	10.39	5.49	108.31	157.61	166.51	80.22
3	6	hay	4.16	7.03	6.24	139.01	194.11	114.76	40.12
4	6	hay	4.04	6.35	4.27	40.71	291.02	121.81	97.58
5	6	hay + carrot	8.52	6.42	92.95	104.61	104.51	88.73	47.66
6	6	hay + carrot	6.08	5.65	110.41	135.81	73.18	112.51	58.41
7	3	no feed	5.39	5.70	264.51	44.99	68.05	44.99	30.75
8	3	no feed	10.24	5.79	54.52	100.91	88.73	114.71	41.32
9	3	hay	4.86	8.98	47.66	187.91	142.41	114.71	29.87
10	3	hay	8.16	5.11	121.81	127.01	28.25	166.51	37.64
11	3	hay + carrot	5.22	4.74	99.22	121.81	65.73	154.61	16.23
12	3	hay + carrot	9.94	4.61	277.12	32.05	57.82	135.81	17.78
13	0	control	8.50	7.24	84.87	121.81	100.91	60.88	33.88
14	0	control	3.42	4.83	99.22	62.20	65.73	145.91	14.62
15	0	control	4.76	7.77	127.01	83.66	77.05	142.41	10.78
16	0	control	6.58	5.69	39.26	87.40	70.52	53.50	18.51
17	0	control	7.22	6.36	44.99	49.32	84.87	92.95	16.59
18	0	control	11.35	6.61	135.81	48.06	30.57	76.05	13.15

**Table 3 animals-14-00664-t003:** Multiple comparisons—effect of the numbers of transportations (3 or 6 times) on the stress level of the rabbits.

(I) Number of Transports	(J) Number of Transports	Mean Difference (I–J)	Std. Error	Sig.	95% Confidence Interval	
					Lower Bound	Upper Bound
Not transported	Three times	−17.65	5.24	0.028 *	−33.08	−2.21
	Six times	−20.30	5.24	0.015 *	−35.73	−4.86
Three times	not transported	17.65	5.24	0.028 *	2.21	33.08
	Six times	−2.64	5.24	0.871	−18.08	12.78
Six times	not transported	20.30	5.24	0.015 *	4.86	35.73
	Three times	2.64	5.24	0.871	−12.78	18.08

Based on observed means. The error term is Mean Square (Error) = 82.382. Where: not transported—rabbits of the control group (*n* = 6) that had not been transported and remained at their housing during the whole examination period, Three times—rabbits that had been transported three times (*n* = 6), Six times—rabbits that had been transported six times (*n* = 6). * The mean difference is significant at the 0.05 level.

## Data Availability

The data presented in this study are available in article.

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
