# Peer review of "The Impact of Transportation on the Cortisol Level of Dwarf Rabbits Bred to Animal-Assisted Interventions"

_animals, 2024, doi:10.3390/ani14050664_

Round 1

Reviewer 1 Report (Previous Reviewer 2)

Comments and Suggestions for Authors

Lines 26-27 are confusing. Consider revising.

Lines 178-179 are confusing. Consider revising

Table 3 is very confusing. Please re-structure this table. It is difficult to tell the 3 day transported rabbit data from the control and 6 day. 

Lines 507-508: this is speculation as to the rabbits' attitudes. Remove this sentence

Instead of labelling your figures and graphs 1-7, list "pre" where you have the "1" and then 1-6 for the days of transport. This will make your key easer to read. 

Did you take samples from the 3 day rabbits on days 4,5, and 6? If so, where is their data? Need to discuss this in the discussion if their cortisol increased like the controls or like the 6 day. Looks like in figure 5 that it went up as well even though they were not transported those days. 

Comments on the Quality of English Language

The English is much better than the original submission, but there are still edits that need to be made throughout the manuscript for sentence structure. 

Author Response

Review 1

Lines 26-27 are confusing. Consider revising.

Whereas the needs and the advantages of people involved in AAI are becoming more and more evident, the needs of animals are not clearly defined, therefore it is a great field of inquiry.

Lines 178-179 are confusing. Consider revising

The sympathetic-adrenal medullary (SAM) and the hypothalamic-pituitary-adrenal cortex (HPA) systems – the adrenal axes – are activated by environmental stimulus. Mammals produces hormones (catecholamines and glucocorticoids) in order to ensure the body to get readily available energy in an emergency action. By monitoring catecholamines and glucocorticoids the animals’ received short-term stress is assessable.

Table 3 is very confusing. Please re-structure this table. It is difficult to tell the 3 day transported rabbit data from the control and 6 day. 

I renamed the table 3 colums and raws.

Lines 507-508: this is speculation as to the rabbits' attitudes. Remove this sentence

The sentence of

“Therefore, the rabbits’ attitude towards the owner was delightful due to proper handling techniques.” is removed.

But I must admit that the housing system of our rabbits are more as hobby “house” rabbits not like stock rabbits. They show more confident behavior against to their owner in our human approach tests than against to people they never met before.

Instead of labelling your figures and graphs 1-7, list "pre" where you have the "1" and then 1-6 for the days of transport. This will make your key easier to read. 

The labelling was renamed.

Did you take samples from the 3 day rabbits on days 4,5, and 6?

If so, where is their data?

Yes we did. All the data is in TABLE 2

Need to discuss this in the discussion if their cortisol increased like the controls or like the 6 day. Looks like in figure 5 that it went up as well even though they were not transported those days. 

The cortisol level increased also in the control animals’ faecal samples. But as our rabbits’ housing system and the care of the animals are more likely as pet animals than stock rabbits

they get more stimulus from the environment which may effect their cortisol level. They are regularly cleaned up, touched, they are grazing in the open-air hutch which all may influence their cortisol level. But the point is that those rabbits who were transported show higher cortisol level in the examined period.

The article was edited and modifications were fulfilled.

LINE 442

In order to get information about the rabbits start up stress level, sampling from all the rabbits had been carried out two days before the transportation started. The total mean of faeces cortisol was 6.71 μg/g at the beginning and it was 6.58 μg/g after the first transport. After a day off (rest period) the examination was continued, the rabbits were taken to the same session and the faeces samples were collected again. Their faeces cortisol concentration rose to 90.85 μg/g.

The control rabbits’ cortisol level also rose up from the second day of the transportation and can be observed that it is similarly move together with the transported rabbits. This tendency is persisted till the end of the research, although significant difference is found to exist between the animals who remained at their housing during the whole study and the transported ones. The reason of the control rabbits’ cortisol rise had been investigated. The circumstances of housing did not change during the study. All the animals were kept individually since they were 12 weeks old. Their cage was placed in the rabbit stable and had not been moved or changed its position for at least 6 weeks before the study had been carried out. No equipment was changed and the daily routine was made exactly the same way as the animals were getting used to previously. Ad libitum hay, pellet food and water were available for them just as before the study. Although it had to be noted, that the rabbits’ housing system and the care of the animals are more likely as pet animals than stock rabbits. They get more stimulus from the environment: they are cleaned up every second day, often touched by their owner, and regularly grazing in the open-air hutch. The animals were taken care by their owner who was handling them during the transportation and was present entirely and they were kept under thermoneutral conditions. Before the experimental period the attendance of the rabbits’ owner meant a positive factor for the animals as they were treated carefully and offered feed as a daily treatment. In contrast, as the owner undertook some of the animals to a stressful event, this attitude toward the owner might have been changed. Because of the significant rise on faeces cortisol concentration in both groups (the control and the transported) we suppose that animals may share their stressful experiences but in a different way as human do, in a way that human senses are not able to express or detect. Previous studies proved that in different stressful situations lambs showed coherence emotional reactivity by alerting their behavioral and physiological responses [55]. Stressful events cause catecholamine and glucocorticoid production throw the activation of the sympathetic system and hypothalamic-pituitary-adrenalin axis [56]. However, there is little known about how stressful experience influence the animals’ relationship with each other, but it is well established that humans’ develop stronger.

To survive the cooperation between the animals may improve social interoperability and animals are to share negative experiences [57]. In our present study the control group’s faeces cortisol level rose up from the second day of transportation (26th of October) till the last occasion (4th of November) when it reduced by more than 80% and established at level 17,92 μg/g for the last examined day. Those rabbits that had not been transported and remained at their housing during the whole study, also showed increased cortisol level.

On the bases of presented data the conclusion is drawn that one occasion of 30 minutes road transport in individual boxes under thermoneutral condition did not make an essential impact on rabbits and as there is no significant difference between the transported and the control groups, we suppose that the animals were able to cope with the stress caused by the first transport. From the second transportation the difference did appear between the transported and control group’s faeces cortisol level although both groups’ stress hormone values rose up. There is a significant difference between the control animals and the three times (p=0.028) transported ones and also between the six times transported (p=0.015) but there is no significant difference between the three and six times transported ones (p=0.871). All animals’ faeces cortisol levels considerably reduced for the last examined day.

We can conclude that transportation negatively affects the rabbits’ anxiety level but based on the large-scale reduction appeared by the 6th session we assume that rabbits might be trained for short term transport, they can get used to it by regularly undertaking them to the procedures while circumspectly handling is ensured. To avoid the exhaustion of the rabbits that are used for assisted/therapy purpose we suggest that trainers who are responsible for the animals and ensure their welfare during the interventions, must take into consideration that transportation itself causes a high stressful experience. Presumably rabbits can get used to the transportations because the faeces cortisol hormone appearance showed remarkable degradation by the last sampling date in our study, but the two-week period training was not enough for the rabbits, therefore future studies are needed.

Reviewer 2 Report (New Reviewer)

Comments and Suggestions for Authors

This is a very interesting study with practical soundness. Unfortunately, the current presentation of the paper is very questionable. The introduction is much too long. I recommend to edit and shorten it. Every reader is aware about the role of pets, the importance of pet assisted therapy and that transport is a stressing event to most species. A more focused of all those questions would improve the paper.

There is a confusion between sections of the paper. Some results are presented in the Mat and Meths section. Table 1 is obviously a result, please, place it in section results. Some comments to clarify some results, are presented in the sections results and discussion, please, make clear sections. In the Mat and meths section, the description of the conditions of life of the animals, is much too long and difficult to read. The paper would gain in clarity with brief descriptions like Diet: ...., Housing: ....

The method for dosification of cortisol, must be shorten and clarified too. Some information is missing and especially information about the ELISA kit would be of high interest. A reference to confirm that this kit is validated for the dosification of fecal cortisol in this species, is required.

The paragraph about Statistics mustbe improved. The approach is very unclear and does not help in understanding the results. There is a spelling mistake on line 365/366 "... as as between...".

In the section Results, the Tables must be improved. The long comments for many lines, explaining what is measured on each day, just prove that the Mat and Meths must be re-written. There is a spelling error on Table 3 title with "comaprison" instead "comparison".

The Discussion must be more structured. Some statements are raising questions. It is especially the case in lines 481-483, when the authors declare that an increase of cortisol means a decrease in welfare. It is very well documented that an increase in cortisol is just associated to emotional activation, that could be positive or negative.

All the results must be in the Results section and not presented after the references. 

Comments on the Quality of English Language

The English is good, despite some minor spelling.

Author Response

This is a very interesting study with practical soundness. Unfortunately, the current presentation of the paper is very questionable.

The introduction is much too long. I recommend to edit and shorten it. Every reader is aware about the role of pets, the importance of pet assisted therapy and that transport is a stressing event to most species. A more focused of all those questions would improve the paper.

The introduction is shortened end edited.

There is a confusion between sections of the paper. Some results are presented in the Mat and Meths section.

Table 1 is obviously a result, please, place it in section results.

Table 1 had been edited (I left the rabbits body weight out of it) so I believe that the Table 1 is now in good session.

 Some comments to clarify some results, are presented in the sections results and discussion, please, make clear sections.

In the Mat and meths section, the description of the conditions of life of the animals, is much too long and difficult to read.

The paper would gain in clarity with brief descriptions like Diet: ...., Housing: ....

I shorted it, and deleted from the text the marked parts:

In the rabbit shed each rabbit was housed individually. The Hungarian and EU legislation in force Decree of the Ministry of Agriculture (32/1999. /III. 31./ and 178/2009. /XII. 29./) in the „General rules for rabbit does and bucks, and suckling and growing rabbits” gives a clear policy: The rabbits have to be housed individually after 12 weeks of age - with the exception of fattening rabbits. The individual cages of our rabbits are placed next to each other and the animals are in visual contact. The pet rabbit cages (95 x 57 x 46 cm) have a corrosion resistant metal coated wire mesh (on the top) and a dark plastic bottom. The metal wire mash and plastic bottom could be separated from each other easily to clean the cage properly and easily every day. The spacious rabbit cage provided all the necessary things rabbits need in their cage like a feeder, a hay container, a hanging water bottle, a separate sleeping area and a rabbit litter box filled with wood pellets (compressed wood shavings and sawdust from untreated wood). To be a house-trained animal rabbits used the litter box to urinate and defecate. All the accessories were made of plastic except the stainless-steel feeder.

As for feeding, the rabbits received a complete diet (Versele-laga complete cuni adult rabbit) that is an all-in-one pellet to avoid selective feeding behavior and to ensure consumption of all the essential nutrients for optimal health. This feed does not contain a coccidiostat or other additives. The analytical data of the feed was as follows: protein 14.0%, fat content 3.0%, crude fibre 20.0%, crude ash 7.0%, Calcium 0.6%, Phosphorus 0.4%. Ingredients: derivatives of vegetable origin (timothy 10% grasses and herbs), Vegetable protein extracts, Vegetables (carrot 4%), Seeds (linseed 2%), Minerals, Fructo-oligosaccharides (0.3%) Marigold, Yucca. Nutritional additives: Vitamin A 10000 IU, Vitamin D3 1200 IU, Vitamin E 80 mg, Vitamin C 100 mg, E1 (iron) 100 mg, E2 (iodine) 2 mg, E4 (copper) 10 mg, E5 (manganese) 75 mg, E6 (zinc) 70 mg, E8 (selenium) 0.2 mg. Technological additives: Antioxidants. The fiber-rich pellet helped to achieve the optimal intestinal function. Besides the pellets, the rabbits had free access to the water nipple drinker, were given hay ad libitum, daily 35-40g fresh, raw carrot stick and a 30-35g apple pieces, both from organic farming, as a treat and vitamin complementary. Feeding was conducted three times a day: at 6AM pellet, 11.30AM the carrot and apple and at 6PM pellet were given. Rabbits had free access to the hay from the hay container water from the nipple drinker the entire day. The cages were equipped with gnawing sticks and supplementary mineral blocks as environmental enrichment. Although the animals are housed individually to prevent harming and social stress, the cages are placed side-by-side that is enabling them to sniff each other and they stay visual contact. In good weather conditions, rabbits were grazing for at least two hours in their mobile open-air hutches every second day.

Veterinary checkups including physical examination like dental health condition check and assessment of the appearance of Spilopsyllus cuniculi and Psoroptes were conducted on all rabbits prior to the transportation examination. All rabbits were vaccinated against myxomatosis and RHD and none of the them were infected with parasites and zoonosis disease. 

The paper would gain in clarity with brief descriptions like Diet: ...., Housing: ....

The method for dosification of cortisol, must be shorten and clarified too.

Some information is missing and especially information about the ELISA kit would be of high interest. A reference to confirm that this kit is validated for the dosification of fecal cortisol in this species, is required.

The detailed description and validation of the method is written in Benedek et.al.’s publication. That is reference 53 and inserted it to the text.

Benedek, I; Altbcker, V.; Molnár, T. (2021) Stress reactivity near birth affects nest building timing and offspring number and survival in the European rabbit (Oryctolagus cuniculus). Plos One, 2021, 16(1) e0246258.

In addition to the collected samples, additional samples were taken to validate the cortisol measurement method. Ten mothers were exposed to social stress by placing two individuals in a transport box (30 � 30 � 40 cm) for one hour. The animals were removed from the box and immobilized by hand to collect blood samples. Subsequently, 2 ml of blood was taken from the ear vein (needle size 21G) within 2 minutes. The samples were centrifuged at 3,000g for10 min, and plasma was separated and stored at -20 ̊C until further measurement. Twenty-four hours after blood collection, a fecal sample was taken as described above and stored at -20 ̊C.

The protocol for fecal extraction was adapted from previously published methods [29]. After freeze-drying, the samples were ground, homogenized, and mixed thoroughly. 200 mg of dry-feces was then placed in a glass vial and 1.6 ml 80% methanol and 200 μl distilled water were added to extract the hormone metabolites. The vials were capped and vortexed for 30 minutes. Samples were then centrifuged (2450 rpm, 20 min, 4 ̊C) and the supernatant was poured off and stored at– 55 ̊C. At the time of use, samples were dried in a chamber (Binder) and reconstituted with ASB buffer at a 1:1 dilution rate. Unlike the original method [29] where a an EIA with a highly specific antibody to a fecal corticosterone metabolites was used, a RIA method was used for cortisol and progesterone, which were developed for hormone determination in the plasma of food animals using tritium labeled hormones (cortisol and progesterone-1,2,6,7-3H(N)) as well as highly-specific polyclonal antibodies raised against cortisol- 21-HS-BSA and 11αOH progesterone11HS:BSA in rabbits. The validation of the RIA method for the fecal metabolites was performed by determining the relationship between the measured levels in the serum and the feces samples. The binding parameters of the cortisol antibody were the following: specific activity >4 TBq/mmol, the lower limit of detection: 15 fmol, affin- ity: KA = 3.0x1010 l/mol. The reconstituted antiserum binds 45% of 10 000 cpm H3 labelled cortisol. Cross-reactivity of the antiserum can be seen in S2 Table. Intra-assay variation was determined to be < 5 CV%, for both cortisol and the progesterone. Inter-assay variation was 9.63 CV% for cortisol, and 1.68 CV% for progesterone. Sensitivity was 0.433 ng/mg for corti- sol, and 0.199 ng/mg for progesterone. For the purpose of determining comparability between cortisol standards in rabbits and the concentration of cortisol in their feces, a high-concentra- tion fecal sample was diluted in a sequential manner. The relationship between fecal cortisol and the standard concentration curve was determined through linear regression, the correla- tion coefficient was 0.998 (r) and the model for it was based on the y = 0.946x + 4.781 equation.

The paragraph about Statistics must be improved. The approach is very unclear and does not help in understanding the results.

This section has been modified with the purpose of explaining that the repeated measures procedure had to be applied because the subsequent measumeremts of the same animals are not independent.

There is a spelling mistake on line 365/366 "... as as between...".

Done.

In the section Results, the Tables must be improved. The long comments for many lines, explaining what is measured on each day, just prove that the Mat and Meths must be re-written.

Yes, thank you. The long comments below the Tables are replaced to Mat and Meths. (Line 350-360

In groups formed by the numbers of transportation, the appearance of cortisol concentration defined by the metabolites of faeces samples (μg/g) were interpreted per day and presented by the following scheme:

  • PRE – collected 2 days before the transport began (day 22/10),
  • 1 - collected 24 hours after the first transport (day 25/11),
  • 2 - collected 24 hours after the second transport (day 27/11),
  • 3 - collected 24 hours after the third transport (day 29/11),
  • 4 - collected 24 hours after the 4th transport (day 01/11),
  • 5 - collected 24 hours after the 5th transport (day 03/11),
  • 6 - collected 24 hours after the 6th transport (day 05/11).

There is a spelling error on Table 3 title with "comaprison" instead "comparison".

Done

The Discussion must be more structured. Some statements are raising questions. It is especially the case in lines 481-483, when the authors declare that an increase of cortisol means a decrease in welfare.

It is very well documented that an increase in cortisol is just associated to emotional activation, that could be positive or negative.

Deleted sentences

Line 452:

The outcome of this research proved insight into the rabbits’ anxiety level during transportation. Our hypothesis that transportation negatively affects the rabbits’ stress level is confirmed. Although it is failed to be proven that additional feed and treats can relieve the transport stress, although significant reduction in animals’ cortisol level measured at the last day in the experimental period suggest that rabbits may get accustomed to it by regular short-term transport journeys.

Line 458

In order to get information about the rabbits start up stress level, sampling from all the rabbits had been carried out two days before the transportation started. The total mean of faeces cortisol was 6.71 μg/g at the beginning and it was 6.58 μg/g after the first transport. After a day off (rest period) the examination was continued, the rabbits were taken to the same session and the faeces samples were collected again. Their faeces cortisol concentration rose to 90.85 μg/g. It must be noted that after the second occasion, not only the transported rabbits’ faeces cortisol concentration increased more than tenfold, but the control groups’ also, although there was no significant difference between the control and the transported animals’ faeces cortisol concentration level.

All the results must be in the Results section and not presented after the references. 

I have made several modifications

There is an indication for the Appendix A in the results section, but the table does not fit to it due it’s larger format than the standard size.

Line 379-380

The detailed statistical data of elapsed days effect is the subject of Appendix A. The difference is significant (p<0.001) between the following days: Pre–5; 1–4; 1–5 and 5-6.

For further modifications pls clarify your request.

Reviewer 3 Report (New Reviewer)

Comments and Suggestions for Authors

The proposed experiment is interesting and reflects a real need to investigate the effects of transport stress on rabbits. The experiment seems to have been carried out correctly, but, as sometimes happens, the results obtained do not fully correspond to the authors' assumptions (which, of course, is not subject to criticism - this happens often). However, the main conclusion about the influence of transport on the development of stress is questionable when the results show comparable trends in the behavior of cortisol levels between the experimental groups and the control group. The statistically significant difference observed on the fourth day (after the 4th transport (day 01/11)) is also unconvincing, since in the next measurement CRT levels again do not differ statistically between the experimental groups and the control group.

In my opinion, the text should be reformulated and the conclusions described anew

Detailed and minor notes on the text are provided below

159 Buil et all. shouldn't there be a date here or a bracket with a number in references like in the next paragraph? (Herbel et.al. [47])

166 “no visual contact” why this criterium, copied from the experiment with dogs , supposed to be good for the animals? During housing animals stayed in visual contact

TAbl. 1 there is not statistical analysis available for this table. Can you explain why you didn’t perform that kind of analysis of the obtained value?

256 I am not sure if the so detailed data about food analysis is necessary. If so, in the author's opinion, please justify it.

Tabl 2 the table description is not clear. The second column looks like divided on two ( one “Time of”  and the second “Feed”-  it's because of the space between words and capitalization of the words “feed”.  This is not a serious error, and it is easy to verify if you read carefully, but it should be corrected to improve the quality of work. Additionally, it  seems also,  that including the word “carrot” in the table supposed to be  possible ( hay+carrot)

Results

the last samples again had lower CRT levels

485 suggests that rabbits may get accustomed to it by regular short-term transport journeys.

Isn't this conclusion too hasty? - we only have one indication showing a reduced CRT level. We don't know whether it was a long-term effect, a trend or maybe a one-time event (actually difficult to explain). Perhaps the conclusion about the need to examine this effect in a longer experiment would confirm this hypothesis. The drop of CRT was observed also in the control group- to what the control animals were supposed to be accustomed?

494 “there was no significant difference between the control and  the transported animals’ feces cortisol concentration level.” That as you indeed concluded suggest significant rise between the days and not the groups!

In my opinion it is difficult to conclude that the transportation caused the stress if animals not transported also presented similar levels of CRT…

504 “No environmental changes occurred”- in fact maybe that is not a truth : removing some animals from the housing could be the environmental change. Could be the stress generating action to the moved and not moved animals- the effect of separation.

545 “We supposed that different types of feed or treats offered to the rabbits during the 545 transportation reduce their anxiety level.”…” our hypothesis is failed to be proven” If this hypothesis failed the further sentence sounds a little bit strange in this manuscript:

“Therefore, we do suggest offering them additional feed like organic carrot or apple as treats and hay.”

In my opinion, authors should present the results and avoid presenting beliefs which are in contrast with the results even if they are sure that additional food is a good solution- just focus on the results

The conclusion “so our hypothesis must be rejected, but we do suggest further studies” sounds quite enough and well reflects the results obtained in the experiment

577 “We can conclude that repeated transportations negatively affect the rabbits’ stress  level but based on the large-scale reduction appeared by the last, the 6th session we assume  that rabbits might be trained for short term transport”

 I will repeat that this conclusion requires stronger proof because I would not draw such far-reaching conclusions based on one measurement…especially that in next sentences authors concluded that “the two-week period  training was not enough for the rabbits”

Concluding: In my opinion this interesting experiment , properly performed is  concluded with not fully proven conclusions, which, in my opinion, require reformulation.

Author Response

Review 3

The proposed experiment is interesting and reflects a real need to investigate the effects of transport stress on rabbits. The experiment seems to have been carried out correctly, but, as sometimes happens, the results obtained do not fully correspond to the authors' assumptions (which, of course, is not subject to criticism - this happens often). However, the main conclusion about the influence of transport on the development of stress is questionable when the results show comparable trends in the behavior of cortisol levels between the experimental groups and the control group. The statistically significant difference observed on the fourth day (after the 4th transport (day 01/11)) is also unconvincing, since in the next measurement CRT levels again do not differ statistically between the experimental groups and the control group.

In my opinion, the text should be reformulated and the conclusions described anew

Detailed and minor notes on the text are provided below

159 Buil et all. shouldn't there be a date here or a bracket with a number in references like in the next paragraph? (Herbel et.al. [47])

Done

166 “no visual contact” why this criterium, copied from the experiment with dogs , supposed to be good for the animals? During housing animals stayed in visual contact

Sentence deleted: During transports the animals had no visual contact with the other dogs in the car and could not see the outside environment.

Tabl. 1 there is not statistical analysis available for this table. Can you explain why you didn’t perform that kind of analysis of the obtained value?

Table 1 had been edited (I left the rabbits body weight out of it) so I believe that the Table 1 is now in good session and provides general information about the animals.

256 I am not sure if the so detailed data about food analysis is necessary. If so, in the author's opinion, please justify it.

The description is shortened and food analysis data was taken out.

Tabl 2 the table description is not clear.

The second column looks like divided on two ( one “Time of”  and the second “Feed”-  it's because of the space between words and capitalization of the words “feed”.  This is not a serious error, and it is easy to verify if you read carefully, but it should be corrected to improve the quality of work. Additionally, it  seems also,  that including the word “carrot” in the table supposed to be  possible ( hay+carrot)

There were missing lines due to the error of layouts. Correction was fulfilled!

Results

the last samples again had lower CRT levels

485 suggests that rabbits may get accustomed to it by regular short-term transport journeys.

Isn't this conclusion too hasty? - we only have one indication showing a reduced CRT level. We don't know whether it was a long-term effect, a trend or maybe a one-time event (actually difficult to explain).

Deleted sentences

Line 452:

The outcome of this research proved insight into the rabbits’ anxiety level during transportation. Our hypothesis that transportation negatively affects the rabbits’ stress level is confirmed. Although it is failed to be proven that additional feed and treats can relieve the transport stress, although significant reduction in animals’ cortisol level measured at the last day in the experimental period suggest that rabbits may get accustomed to it by regular short-term transport journeys.

Perhaps the conclusion about the need to examine this effect in a longer experiment would confirm this hypothesis. The drop of CRT was observed also in the control group- to what the control animals were supposed to be accustomed?

494 “there was no significant difference between the control and  the transported animals’ feces cortisol concentration level.”

Deleted sentence

Line 458:

In order to get information about the rabbits start up stress level, sampling from all the rabbits had been carried out two days before the transportation started. The total mean of faeces cortisol was 6.71 μg/g at the beginning and it was 6.58 μg/g after the first transport. After a day off (rest period) the examination was continued, the rabbits were taken to the same session and the faeces samples were collected again. Their faeces cortisol concentration rose to 90.85 μg/g. It must be noted that after the second occasion, not only the transported rabbits’ faeces cortisol concentration increased more than tenfold, but the control groups’ also, although there was no significant difference between the control and the transported animals’ faeces cortisol concentration level.

That as you indeed concluded suggest significant rise between the days and not the groups!

In my opinion it is difficult to conclude that the transportation caused the stress if animals not transported also presented similar levels of CRT…

I edited the discussion and I believe that I gave proper answer and explanations.

In order to get information about the rabbits start up stress level, sampling from all the rabbits had been carried out two days before the transportation started. The total mean of faeces cortisol was 6.71 μg/g at the beginning and it was 6.58 μg/g after the first transport. After a day off (rest period) the examination was continued, the rabbits were taken to the same session and the faeces samples were collected again. Their faeces cortisol concentration rose to 90.85 μg/g.

The control rabbits’ cortisol level also rose up from the second day of the transportation and can be observed that it is similarly move together with the transported rabbits. This tendency is persisted till the end of the research, although significant difference is found to exist between the animals who remained at their housing during the whole study and the transported ones. The reason of the control rabbits’ cortisol rise had been investigated. The circumstances of housing did not change during the study. The rabbits were kept under thermoneutral conditions and were individually housed since they were 12 weeks old. Their cage was placed in the rabbit stable and had not been moved or changed its position for at least 6 weeks before the study had been carried out. No equipment was changed and the daily routine was made exactly the same way as the animals were getting used to previously. Ad libitum hay, pellet food and water were available for them just as before the study. Although it had to be noted, that the rabbits’ housing system and the care of the animals are more likely as pet animals than stock rabbits. They get more stimulus from the environment: they are cleaned up every second day, often touched by their owner, and regularly grazing in the open-air hutch. The animals were taken care by their owner who was handling them during the transportation and was present entirely. Before the experimental period the attendance of the rabbits’ owner meant a positive factor for the animals as they were treated carefully and offered feed as a daily treatment. In contrast, as the owner undertook some of the animals to a stressful event, this attitude toward the owner might have been changed. Because of the significant rise on faeces cortisol concentration in both groups (the control and the transported) we suppose that animals may share their stressful experiences but in a different way as human do, in a way that human senses are not able to express or detect. Previous studies proved that in different stressful situations lambs showed coherence emotional reactivity by alerting their behavioral and physiological responses [55]. Stressful events cause catecholamine and glucocorticoid production throw the activation of the sympathetic system and hypothalamic-pituitary-adrenalin axis [56]. However, there is little known about how stressful experience influence the animals’ relationship with each other, but it is well established that humans’ develop stronger.

To survive the cooperation between the animals may improve social interoperability and animals are to share negative experiences [57]. In our present study the control group’s faeces cortisol level rose up from the second day of transportation (26th of October) till the last occasion (4th of November) when it reduced by more than 80% and established at level 17,92 μg/g for the last examined day. Those rabbits that had not been transported and remained at their housing during the whole study, also showed increased cortisol level.

On the bases of presented data the conclusion is drawn that one occasion of 30 minutes road transport in individual boxes under thermoneutral condition did not make an essential impact on rabbits and as there is no significant difference between the transported and the control groups, we suppose that the animals were able to cope with the stress caused by the first transport. From the second transportation the difference did appear between the transported and control group’s faeces cortisol level although both groups’ stress hormone values rose up. There is a significant difference between the control animals and the three times (p=0.028) transported ones and also between the six times transported (p=0.015) but there is no significant difference between the three and six times transported ones (p=0.871). All animals’ faeces cortisol levels considerably reduced for the last examined day.

We can conclude that transportation negatively affects the rabbits’ anxiety level but based on the large-scale reduction appeared by the 6th session we assume that rabbits might be trained for short term transport, they can get used to it by regularly undertaking them to the procedures while circumspectly handling is ensured. To avoid the exhaustion of the rabbits that are used for assisted/therapy purpose we suggest that trainers who are responsible for the animals and ensure their welfare during the interventions, must take into consideration that transportation itself causes a high stressful experience. Presumably rabbits can get used to the transportations because the faeces cortisol hormone appearance showed remarkable degradation by the last sampling date in our study, but the two-week period training was not enough for the rabbits, therefore future studies are needed.

504 “No environmental changes occurred”- in fact maybe that is not a truth : removing some animals from the housing could be the environmental change. Could be the stress generating action to the moved and not moved animals- the effect of separation.

I deleted the part of the sentence, where No environmental changes occurred.

Line: 465

The animals were taken care by their owner who was handling them during the transportation and was present entirely. No environmental changes occurred; the animals were kept under thermoneutral conditions. Before the experimental period the attendance of the rabbits’ owner always meant a joyful time for the animals as they were treated carefully and offered feed as a daily treatment.

But I have to note that the housing system of our rabbits are more as hobby “house” rabbits not like stock rabbits. They show more confident behavior against to their owner in our human approach tests. They are bred for to participate animal assisted interventions and they get used to environmental changes like we take out them from their cage. They regularly free gazing in the garden for example. They are exposed to different stimulations which is caused by the “pet animal” lifestyle.  

545 “We supposed that different types of feed or treats offered to the rabbits during the 545 transportation reduce their anxiety level.”…” our hypothesis is failed to be proven” If this hypothesis failed the further sentence sounds a little bit strange in this manuscript:

“Therefore, we do suggest offering them additional feed like organic carrot or apple as treats and hay.”

In my opinion, authors should present the results and avoid presenting beliefs which are in contrast with the results even if they are sure that additional food is a good solution- just focus on the results

Yes, thank you for your note! Sentence removed.

… our hypothesis must be rejected, but we do suggest further studies to investigate whether the anxiety of the animals is reducible by feed as it had been observed that some of the rabbits ate while they were at their carrier cages (Figure 3 (a) and (b). Therefore, we do suggest offering them additional feed like organic carrot or apple as treats and hay.

577 “We can conclude that repeated transportations negatively affect the rabbits’ stress  level but based on the large-scale reduction appeared by the last, the 6th session we assume  that rabbits might be trained for short term transport”

 I will repeat that this conclusion requires stronger proof because I would not draw such far-reaching conclusions based on one measurement…especially that in next sentences authors concluded that “the two-week period  training was not enough for the rabbits”

The part of the sentence removed.

We can conclude that repeated transportations negatively affect the rabbits’ stress level. but based on the large-scale reduction appeared by the last, the 6th session we assume that rabbits might be trained for short term transport. To avoid the exhaustion of the rabbits that is used for assisted/therapy purpose we suggest that trainers who are responsible for the animals and ensure their welfare during the interventions, must take into consideration that transportation itself causes a high stressful experience. Presumably rabbits can get used to the transportations because the faeces cortisol hormone appearance showed a significant degradation by the last sampling date in our study, but the two-week period training was not enough for the rabbits, therefore future studies are needed.

Concluding: In my opinion this interesting experiment, properly performed is  concluded with not fully proven conclusions, which, in my opinion, require reformulation.

Round 2

Reviewer 2 Report (New Reviewer)

Comments and Suggestions for Authors

Thank you for re-organizing the paper, it makes it easier to read. I still consider that the introduction is too long, but it seems to be your will to keep it this way. I respect it.

The section Results is more organized.

Reviewer 3 Report (New Reviewer)

Comments and Suggestions for Authors

The authors responded to all my comments and the manuscript was significantly improved. In my opinion, it can be published in its current form

This manuscript is a resubmission of an earlier submission. The following is a list of the peer review reports and author responses from that submission.

Round 1

Reviewer 1 Report

Comments and Suggestions for Authors

The Authors have investigated an interesting topic and provided a relevant background for their research. However, objectives of the study are missing, they should be clearly defined. Only hypothesis is presented at the end of Introduction.

Major methodological concerns:

The Authors declare that the gender effect on transportation stress was not taken into consideration. Reasoning for this decision should be included.

The rabbits under study were housed individually despite welfare recommendations. Did the authors consider the impact of individual housing on stress levels? Changes in GCM reflect chronic stress rather than effect of acute stress such as short transportation.

Only dates of transport and dates of collecting samples are given. It is not clear how the time of transport and time of collecting samples correspond. How long after the rabbits were exposed to transport stress were the samples collected? There is a species-specific delay before the stress-induced change in fecal cortisol metabolites can be seen.

No information on analysis of feces is given in the manuscript beside: „.. the laboratory examination had been carried out and GCM had been determined. The rabbits’ faeces cortisol level was analyzed…“ which is very confusing. Did the Authors analyze glucocorticoid metabolites or concentration of cortisol in feces? Given the common practice in this type of studies I would expect the former. However, the result section presents cortisol levels…

Given the methodological questions I am unable to properly assess discussion and conclusions.

The careful proof-reading is recommended to eliminate typos and to unify formatting of the manuscript.

Author Response

Blue reviewer:

The Authors have investigated an interesting topic and provided a relevant background for their research.

  1. However, objectives of the study are missing, they should be clearly defined. Only hypothesis is presented at the end of Introduction.

Objectives are added, Line: 197-202

Rabbits’ participation in Animal Assisted Interventions is an increasing area, however no recommendation is available about how to use them while animal welfare is guaranteed. As those rabbits that are service in AAI need to be delivered to the plot of the sessions, our present study’s main objective was to investigate the effect of the repeated transportation to the rabbits anxiety level and give general recommendations about how to fulfill animal welfare requirement during the procedures.

Major methodological concerns:

  1. The Authors declare that the gender effect on transportation stress was not taken into consideration. Reasoning for this decision should be included.

„...the gender effect on transportation stress was not taken into consideration.”  (lines 2011-212).

Our main goal was to investigate the repeated transportation effects on the rabbits in general. Our does and bucks are selected for tameness during seven generations. According to our knowledge previously available study or data that would give a recommendation regarding the sex of the rabbit used for AAI is not provided. Therefore, the focus of our interest was the reaction of tame rabbits’ for the procedures of transportation regardless of their gender.

Michaela Součková et al. in Applied Animal Behaviour Science, 2023, volume 262, article title: Behavioural reactions of rabbits during AAI sessions, writes that “there is only one study so far assessing rabbits stress during an AAI session (Suba-Bokodi et al., 2022)

In our previous study (Changes in the Stress Tolerance of Dwarf Rabbits in Animal-Assisted Interventions Appl. Sci. 202212(14), 6979; https://doi.org/10.3390/app12146979) we also did not take into consideration the effect of gender to the AAI work as there are pros and cons of each gender:

  • according to d’Ovidio et al.’s online questionnaire (n = 634), the chance of aggressive behavior of pet rabbits against their owner or a strange person is significantly lower in intact males than in neutered males and does, and buck rabbits’ interest in their owner is significantly higher [D’Ovidio, D.; Pierantoni, L.; Noviello, E.; Pirrone, F. Sex differences in human-directed social behavior in pet rabbits. Veter.-Behav. 201615, 37–42.]
  • the intact males spray urine to determine their territory [Crowell-Davis, S.L. Behavior Problems in Pet Rabbits. Exot. Pet Med. 200716, 38–44].
  • We did not experience the calmer temperament of the intact males, and because of their urine spraying behavior, we preferred using does for AAI works. [Suba-Bokodi, É.; Nagy, I.; Molnár, M. Changes in the Stress Tolerance of Dwarf Rabbits in Animal-Assisted Interventions Appl. Sci., 12 (2022), p. 6979]

  1. The rabbits under study were housed individually despite welfare recommendations. Did the authors consider the impact of individual housing on stress levels?

The individual cages of our rabbits are placed next to each other and the animals are in visual contact. The Hungarian and EU legislation in force Decree of the Ministry of Agriculture (32/1999. /III. 31./ and 178/2009. /XII. 29./)  in theGeneral rules for rabbit does and bucks, and suckling and growing rabbits” gives a clear policy: The rabbits have to be housed individually after 12 weeks of age - with the exception of fattening rabbits.

Changes in GCM reflect chronic stress rather than effect of acute stress such as short transportation.

  • Pesenhofer et. al. done GCM analysis in cows (n = 207) compared two different devices for restraint during functional claw trimming (acute stressor):

[Pesenhofer G, Palme R, Pesenhofer RM and Kofler J 2006 Comparison of two methods of fixation during functional claw trimming, walk-in crush versus tilt table, in dairy cows using fae- cal cortisol metabolite concentrations and daily milk yield as parameters. Wiener Tierärztliche Monatsschrift 93: 288-294]

  • Monclús, R., Rödel, H.G., Palme, R. et al. Non-invasive measurement of the physiological stress response of wild rabbits to the odour of a predator – acute stress

Chemoecology 16, 25–29 (2006). https://doi.org/10.1007/s00049-005-0324-6

From the Department of Diagnostic Laboratory, University of Veterinary Medicine (Budapest, Hungary) we received the following additional description:

Cortisol levels were measured from the faeces based on their breakdown products, extracted from the faeces samples by adding 4 ml of ethanol for 500 mg of faeces, followed by vortexing for 3 minutes. Then, 10 minutes centrifugation on 2000 rpm was applied. Cortisol-containing supernatant was collected and used as samples. 

The concentration of metabolites was determined with a cortisol ELISA kit (DEH3388, Demeditec GmBH, Kiel, Germany), according to the manufacturer’s protocol. Analytical sensitivity 0.38 ng/ml; range 10-800 ng/ml; intra-assay CV <5%; inter-assay CVs were 5.2 and 7.8% for control 1 and control 2, respectively. Raw cortisol concentration data was translated and expressed as μg/g.

Lines: 323-331

  1. Only dates of transport and dates of collecting samples are given. vb. It is not clear how the time of transport and time of collecting samples correspond.

The exact times are also given:

Line 254.

  1. Materials and methods

2.3 Transportation:

For two weeks every second day rabbits were taken for a 30-minute long transport at 8.30AM.

Line 306-315.

  1. Materials and methods

2.4. Collecting, storing and transporting the faeces samples to laboratory

The samples were collected as follows:

  • The plastic bottom of the animal’s individual cage and the litter box were cleaned and sterilized by a biocidal product with a spectrum of bactericidal, fungicidal and virucidal effects (SteriClean Farm, active ingredient: Sodium hypochlorite solution (0.05%) a daily at 7.30AM.
  • After restoring all equipment in the cage, the litter box was filled with wood pellets again.
  • At 8:30AM if faeces appeared in litter box, they were removed immediately.
  • After 8.30AM from the first faeces of the animals, samples were collected using sterile gloves to each individual. Every 15 minute the conductor of the examination checked the litter boxes and collected the new samples.

  1. How long after the rabbits were exposed to transport stress were the samples collected?

The samples were collected 24 hours after the transportation. Detailed above. 

  1. There is a species-specific delay before the stress-induced change in fecal cortisol metabolites can be seen.

According to Benedek et al. [Benedek, I; Altbcker, V.; Molnár, T. (2021) Stress reactivity near birth affects nest building timing and offspring number and survival in the European rabbit (Oryctolagus cuniculus). Plos One, 2021, 16(1) e0246258.] 24 hours after the pressure of stress the decomposition of hormone cortisol appears on the faeces.

There are several factors that influences faecal glucocorticoid metabolites:

  • In Palme’s publication (Monitoring stress hormone metabolites as a useful, non-invasive tool for welfare assessment in farm animals, Animal Welfare 2012, 21: 331-337 ISSN 0962-7286 doi: 10.7120/09627286.21.3.331) is written: Besides time of collection, the conditions under which the samples are stored are critical, as further bacterial metabolism of the excreted FCM has been reported (Morrow et al 2002; Möstl et al 2005; Lexen et al 2008). Thus, it is recommended to collect fresh faecal samples and freeze them immediately (< 30 min) after defaecation. Storing faeces in a transportable ice box before transferring them into a deep freezer may also help reduce possible metabolism by bacterial enzymes. Keeping samples frozen (–20°C) until analysis is necessary.
  • There is a time delay between increased plasma GC levels and their reflection in the excreted FCM (gut passage time from the duodenum to the rectum; Palme et al 1996). As a consequence, faecal samples offer the advantage of a post hoc evaluation (Touma & Palme 2005). This time delay is species dependent but may be influenced by the indi- vidual and other factors such as feed intake (Morrow et al 2002)

Therefore, we performed the sampling by the same timing after every transportation. All the circumstances were standardized: they take the same feed, kept in thermoneutral condition etc. We ensured that every samples were collected freshly (checking the animals in every 15 minutes and took, labeled froze down the fresh faecal samples immediately).

I accept that we might not get the peak of the faeces metabolites concentration but we do get data of its tendency that is an evaluable information.

Besides time of collection, the conditions under which

the samples are stored are critical, as further bacterial

metabolism of the excreted FCM has been reported

(Morrow et al 2002; Möstl et al 2005; Lexen et al 2008).

Thus, it is recommended to collect fresh faecal samples

and freeze them immediately (< 30 min) after defaeca-

tion. Storing faeces in a transportable ice box before

transferring them into a deep freezer may also help

reduce possible metabolism by bacterial enzymes.

Keeping samples frozen (–20°C) until analysis is

necessary.

Besides time of collection, the conditions under which

the samples are stored are critical, as further bacterial

metabolism of the excreted FCM has been reported

(Morrow et al 2002; Möstl et al 2005; Lexen et al 2008).

Thus, it is recommended to collect fresh faecal samples

and freeze them immediately (< 30 min) after defaeca-

tion. Storing faeces in a transportable ice box before

transferring them into a deep freezer may also help

reduce possible metabolism by bacterial enzymes.

Keeping samples frozen (–20°C) until analysis is

necessary.

Besides time of collection, the conditions under which

the samples are stored are critical, as further bacterial

metabolism of the excreted FCM has been reported

(Morrow et al 2002; Möstl et al 2005; Lexen et al 2008).

Thus, it is recommended to collect fresh faecal samples

and freeze them immediately (< 30 min) after defaeca-

tion. Storing faeces in a transportable ice box before

transferring them into a deep freezer may also help

reduce possible metabolism by bacterial enzymes.

Keeping samples frozen (–20°C) until analysis is

necessary

  1. No information on analysis of feces is given.

We completed the it in 2. Materials and methods, 2.4. Collecting, storing and transporting the faeces samples to laboratory

Cortisol levels were measured from the faeces based on their breakdown products, extracted from the faeces samples by adding 4 ml of ethanol for 500 mg of faeces, followed by vortexing for 3 minutes. Then, 10 minutes centrifugation on 2000 rpm was applied. Cortisol-containing supernatant was collected and used as samples. 

The concentration of metabolites was determined with a cortisol ELISA kit (DEH3388, Demeditec GmBH, Kiel, Germany), according to the manufacturer’s protocol. Analytical sensitivity 0.38 ng/ml; range 10-800 ng/ml; intra-assay CV <5%; inter-assay CVs were 5.2 and 7.8% for control 1 and control 2, respectively. Raw cortisol concentration data was translated and expressed as μg/g.

Lines: 323-331

  1. In the manuscript beside: „.. the laboratory examination had been carried out and GCM had been determined. The rabbits’ faeces cortisol level was analyzed…“ which is very confusing. Did the Authors analyze glucocorticoid metabolites or concentration of cortisol in feces?

Yes, thank you for the remark: We made a correction in the whole article

Cortisol levels were measured from the faeces based on their breakdown products, extracted from the faeces samples.

Reviewer 2 Report

Comments and Suggestions for Authors

Overall, a good study that needs to be published and this special issue seems to be the right place for it. However, the figures and graphs are confusing and need to be re-worked. The materials and methods needs a little more information about the group G animals and timing of sampling in relation to the transport.

In Table 1, add M for male and F for female instead of "1" and "2"

Line 252-254. Were they transported every other day or Monday, Wed, and Friday? Clarify

For table 2, Better to label data before and after transport rather than dates. For example - "Pre" Day 1, day 3, day 5, etc. When were the samples taken in relation to transport (days or hours). What are the "*" for?

Were the group G rabbits loaded into carriers or were they allowed to remain in their cages? Need more information about this group

Table 3 is confusing and probably not needed unless you can better justify what it is showing. Here you state the "estimate" for the group G rabbits is 49.8, but in Figure 1, you list 51.73.

Please add letters or symbols to indicate significance in all figures

Line 384- P values were below or above significance? should be above

Lines 407-415 are confusing in how you have the data presented. I like the percentage of difference being presented, but it is difficult to follow. Present the change in each per day compared to the control. 

Line 449-450: Take this sentence out. Speculation as to the animals' feelings.

Why were some of the rabbits only transported 3 times? Your data would be stronger with more animals in each group

Appendix A and B are also difficult to follow. Instead of listing numbers, list the group letter. What is the "estimate" measuring? If cortisol, label it as cortisol

The abstract has multiple typo errors that need to be corrected. 

There is a concern that there were only really 2 animals per group which is too low. It would have been stronger if you had more per group (by transporting all the animals 6 times). Please acknowledge this in the discussion section that it may be the reason significance was not reached. 

Comments on the Quality of English Language

Several errors in the abstract, needs some tweeking of sentences for clarity. 

Author Response

Green comments

Overall, a good study that needs to be published and this special issue seems to be the right place for it.

However, the figures and graphs are confusing and need to be re-worked.

  1. The materials and methods needs a little more information about the group G animals

Extra information added at Line 294: 

remained at heir housing during the whole examination period.

  1. and timing of sampling in relation to the transport.

The exact times are given:

Line 254.

  1. Materials and methods

2.3 Transportation:

For two weeks every second day rabbits were taken for a 30-minute long transport at 8.30AM.

Line 306-315.

  1. Materials and methods

2.4. Collecting, storing and transporting the faeces samples to laboratory

The samples were collected as follows:

  • The plastic bottom of the animal’s individual cage and the litter box were cleaned and sterilized by a biocidal product with a spectrum of bactericidal, fungicidal and virucidal effects (SteriClean Farm, active ingredient: Sodium hypochlorite solution (0.05%) a daily at 7.30AM.
  • After restoring all equipment in the cage, the litter box was filled with wood pellets again.
  • At 8:30AM if faeces appeared in litter box, they were removed immediately.
  • After 8.30AM from the first faeces of the animals, samples were collected using sterile gloves to each individual. Every 15 minute the conductor of the examination checked the litter boxes and collected the new samples.

According to Benedek et al. [Benedek, I; Altbcker, V.; Molnár, T. (2021) Stress reactivity near birth affects nest building timing and offspring number and survival in the European rabbit (Oryctolagus cuniculus). Plos One, 2021, 16(1) e0246258.] 24 hours after the pressure of stress the decomposition of hormone cortisol appears on the faeces.

But There are several factors that influences faecal glucocorticoid metabolites:

  • In Palme’s publication (Monitoring stress hormone metabolites as a useful, non-invasive tool for welfare assessment in farm animals, Animal Welfare 2012, 21: 331-337 ISSN 0962-7286 doi: 10.7120/09627286.21.3.331) is written: Besides time of collection, the conditions under which the samples are stored are critical, as further bacterial metabolism of the excreted FCM has been reported (Morrow et al 2002; Möstl et al 2005; Lexen et al 2008). Thus, it is recommended to collect fresh faecal samples and freeze them immediately (< 30 min) after defaecation. Storing faeces in a transportable ice box before transferring them into a deep freezer may also help reduce possible metabolism by bacterial enzymes. Keeping samples frozen (–20°C) until analysis is necessary.
  • There is a time delay between increased plasma GC levels and their reflection in the excreted FCM (gut passage time from the duodenum to the rectum; Palme et al 1996). As a consequence, faecal samples offer the advantage of a post hoc evaluation (Touma & Palme 2005). This time delay is species dependent but may be influenced by the indi- vidual and other factors such as feed intake (Morrow et al 2002)

Therefore, we performed the sampling by the same timing after every transportation. All the circumstances were standardized: the animals took the same feed, kept in thermoneutral condition etc. We ensured that every samples were collected freshly (checking the animals in every 15 minutes and took, labeled froze down the fresh faecal samples immediately).

  1. In Table 1, add M for male and F for female instead of "1" and "2"

Done – Table 1. column 2 Gender.

  1. Were they transported every other day or Monday, Wed, and Friday? Clarify

More detailed description was added:

... is working three days per week in AAI and need to be transported to the intervention’s spot. During the examined period the rabbits were transported on Monday, Wednesday and Friday, while all the other days they remained at their housing.

Line 256-259

  1. For table 2, Better to label data before and after transport rather than dates. For example - "Pre" Day 1, day 3, day 5, etc. When were the samples taken in relation to transport (days or hours).

Done, and more detailed from line Line 330 - 337

  1. What are the "*" for?

By mistake, Deleted!

  1. Were the group G rabbits loaded into carriers or were they allowed to remain in their cages? Need more information about this group

They remained in their cages, nothing happened to them.

The information added, Line 365

  1. Table 3 is confusing and probably not needed unless you can better justify what it is showing.

We removed table 3.

  1. Please add letters or symbols to indicate significance in all figures

Done: Line 415-420

  1. Line 384- P values were below or above significance? should be above

Corrected (p>0.05).

  1. Lines 407-415 are confusing in how you have the data presented. I like the percentage of difference being presented, but it is difficult to follow. Present the change in each per day compared to the control.

For the first occasion the transported rabbits’ cortisol level was 4,89% higher than the control group’s and the difference decreased by the second occasion. The tendency of getting higher gap between the transported animals and those rabbits who remained at their housing can be determined but none of the days’ difference is significant. For the third occasion the control rabbits’ cortisol mean decreased from 88.53 μg/g to 69.22 μg/g while the transported ones increased by 12.68 μg/g.

Line: 456-461

  1. Line 449-450: Take this sentence out. Speculation as to the animals' feelings.

Done

  1. Why were some of the rabbits only transported 3 times?

Done: Line 280-289

Basically the 18 rabbits were divided into three groups:

  • 6 of them were transported 6 times to examine prolonged period stress effects on the animals’,
  • 6 of them were transported 3 times to examine intermediate long period stress effects on the animals’ and
  • 6 animals formed the control group that did not transported any times and remained at their housing during the whole examination period.

Your data would be stronger with more animals in each group

Our animals are not commercially available ones. All of our rabbits are selected for tameness during seven generations and bred for serving in Animal Assisted Interventions. Their ancestors are successfully serving in primary schools, kindergartens and also in elderly taking care homes.

  1. Appendix A and B are also difficult to follow. Instead of listing numbers, list the group letter.

Done Appendix A and B, Line: 579 and 591

What is the "estimate" measuring? If cortisol, label it as cortisol

It measures the differences among the cortisol LS means of the different groups

  1. The abstract has multiple typo errors that need to be corrected.

Done

  1. There is a concern that there were only really 2 animals per group which is too low. It would have been stronger if you had more per group (by transporting all the animals 6 times). Please acknowledge this in the discussion section that it may be the reason significance was not reached.

Done: 533-539

Round 2

Reviewer 1 Report

Comments and Suggestions for Authors

The authors have revised the manuscript to address most remarks. I am still not convinced about their methodology or its description (it is more common and reasonable to measure GC metabolites in feces than cortisol levels) and the findings are very limited. However, there is a lack of studies on aspects affecting welfare of animals used in animal assisted interventions other than dogs and this study might be a starting point for future research thus, I recommend its publication.